# A Retrospective, Monocentric Study Comparing Co and Secondary Infections in Critically Ill COVID-19 and Influenza Patients

**DOI:** 10.3390/antibiotics11060704

**Published:** 2022-05-24

**Authors:** Diane Marcoux, Isabelle Etienne, Alain Van Muylem, Elisa Gouvea Bogossian, Nicolas Yin, Fabio Silvio Taccone, Maya Hites

**Affiliations:** 1Clinic of Infectious Diseases, HUB-Erasme Hospital, 1070 Brussels, Belgium; diane.marcoux@chu-charleroi.be; 2Department of Pneumology, HUB-Erasme Hospital, 1070 Brussels, Belgium; isabelle.etienne@erasme.ulb.ac.be (I.E.); alain.van.muylem@erasme.ulb.ac.be (A.V.M.); 3Department of Intensive Care, HUB-Erasme Hospital, 1070 Brussels, Belgium; elisagobog@gmial.com (E.G.B.); fabio.taccone@erasme.ulb.ac.be (F.S.T.); 4Department of Microbiology, Laboratoire Hospitalier Universitaire de Bruxelles-Universitair Laboratorium Brussel (LHUB-ULB), Université Libre de Bruxelles, 1070 Brussels, Belgium; nicolas.yin@erasme.ulb.ac.be

**Keywords:** COVID-19, Influenza, co-infection, secondary infections, aspergillosis

## Abstract

Few data are available on infectious complications in critically ill patients with different viral infections. We performed a retrospective monocentric study including all of the patients admitted to the intensive care unit (ICU) with confirmed COVID-19 (as of 13 March 2020) or Influenza A and/or B infections (as of 1 January 2015) until 20 April 2020. Coinfection and secondary infections (occurring within and after 48 h from admission, respectively) were recorded. Fifty-seven COVID-19 and 55 Influenza patients were included. Co-infections were documented in 13/57 (23%) COVID-19 patients vs. 40/55 (73%) Influenza patients (*p* < 0.001), most of them being respiratory (9/13, 69% vs. 35/40, 88%; *p* = 0.13) and of bacterial origin (12/13, 92% vs. 29/40, 73%; *p* = 0.25). Invasive aspergillosis infections were observed only in Influenza patients (8/55, 15%). The COVID-19 and Influenza patients presented 1 (0–4) vs. 0 (0–4) secondary infections (*p* = 0.022), with comparable sites being affected (lungs: 35/61, 57% vs. 13/31, 42%; *p* = 0.16) and causative pathogens occurring (Gram-negative bacteria: 51/61, 84% vs. 23/31, 74%; *p* > 0.99). The COVID-19 patients had longer ICU lengths of stay (15 (–65) vs. 5 (1–89) days; *p* = 0.001), yet the two groups had comparable mortality rates (20/57, 35% vs. 23/55, 41%; *p* = 0.46). We report fewer co-infections but more secondary infections in the ICU COVID-19 patients compared to the Influenza patients. Most of the infectious complications were respiratory and of bacterial origin.

## 1. Introduction

In December 2019, a new virus named Severe Acute Respiratory Syndrome Coronavirus 2 (SARS-CoV-2) was identified in Wuhan city, Hubei Province, China, causing a new disease called COVID-19, which can lead to severe acute respiratory distress syndrome (ARDS) with a relatively high risk of death [1]. This virus quickly spread worldwide, resulting in a global health crisis and a rapid saturation of health care services, including intensive care units (ICU) [2].

Several studies have already been published concerning the occurrence of co-infections and/or secondary infections in ICU COVID-19 patients [3,4,5,6,7,8,9,10,11]. Most of these studies are retrospective and report data on small cohorts including heterogenous populations. There are only two published meta-analyses based on retrospective studies: one that included small cohorts from six studies only focusing on ICU patients and eight studies focusing on both ICU and non-ICU patients that reported on COVID-19 co-infections [11]; and one that included five studies focusing only on ICU patients that described co-infections (four of those studies) and secondary infections (one of those studies) [3]. According to these studies, the rate of co-infections varied from 4.3% to 28% [3,4,5,6,7,8,9], and the rate of secondary infections ranged from 40.7% to 51% [4,7,8,9,10]. It is unclear whether the risk of coinfections or secondary infections is similar to that of other viral infections.

Influenza affects up to 20% of the population each year. Bacterial and fungal co-infections are frequent occurrences among ICU patients with Influenza infection. Co-infected patients have higher morbidity and mortality rates than those without co-infections [12]. The mechanisms of co-pathogenesis have been studied and are multifactorial [12,13]. First, the virus causes epithelial dysfunction. The ciliary damage prevents efficient pathogen clearance, and the epithelial cells and surfactant destruction give the pathogens access to nutrients. Second, these structural damages, together with the local inflammation, facilitate bacterial adherence to their specific receptors. Bacteria also produce pro-inflammatory cytokines resulting in the synergistic activation of immune response. Third, the immune response to the influenza virus interferes with the normal pathogen recognition effector responses during and after the viral infection, enabling secondary bacterial infections [14].

A large French retrospective study comparing 89,530 COVID-19 patients to 45,819 Influenza patients reported a higher in-hospital death rate among the COVID-19 patients requiring ICU admission compared to the Influenza patients; however, the rates of coinfections and secondary infections were not reported [15]. Another retrospective study comparing 642 COVID-19 patients to 742 Influenza patients reported more secondary bacterial infections in the COVID-19 patients, which was an independent predictor of death in the COVID-19 patients [16]. In a retrospective study, Bardi and al. also reported a higher ICU mortality in secondary infected COVID-19 ICU patients compared to those not infected [9].

This study therefore aims to describe whether the occurrence, type, and outcome of co-infections and secondary infections in ICU patients are different between the COVID-19 and Influenza diseases.

## 2. Results

### 2.1. Patients’ Characteristics

A total of 57 COVID-19 patients and 55 Influenza A or B patients were eligible for the analysis over the study period. The characteristics of the study population are presented in Table 1.

The patients were predominantly men in both groups and had comparable ages (63 years (53–74)), Charlson Comorbidity Indexes (4 (2–5)), and prevalence of obesity (32%). The COVID-19 patients were less frequently smokers and had less chronic cardiomyopathy, pulmonary disease, and kidney failure than the Influenza patients. They were also significantly less immunocompromised than the Influenza patients, receiving chronic steroid treatment less frequently.

### 2.2. Clinical Data upon ICU Admission

The clinical data upon ICU admission are presented in Table 2a. The patients were mainly admitted to the ICU for respiratory failure. The patients were severely ill, with comparable baseline SOFA and SAPS3 scores and PaO_2_/FiO_2_ ratios in both patient groups (medians of 6 (3–10); 58 (48–68) and 145 (106–213), respectively). The delay between the hospital and ICU admission was significantly shorter in the COVID-19 group compared to that in the Influenza group. Upon admission, the COVID-19 patients presented significantly more ground glass features on the CT-scan and a lower leucocyte count, but they presented a higher lymphocyte count and higher c-reactive protein and lactate dehydrogenase values than the Influenza patients. Significantly fewer COVID-19 patients received antibiotics before and during the first 48 h of ICU admission compared to the Influenza cohort.

### 2.3. Co-Infections

The data on co-infections are reported in Table 2b and Figure 1. The microbiological documentation was more systematic in the COVID-19 patients than in the Influenza patients upon ICU admission, meaning that significantly more BAL, blood, and urine cultures were performed for the COVID-19 patients compared to the Influenza patients. Nevertheless, the COVID-19 patients presented significantly fewer co-infections than the Influenza patients. The co-infections were predominantly respiratory in both groups. The most frequent pathogens were Gram-positive cocci (*Streptococcus pneumoniae* in the COVID-19 group and *Staphylococcus aureus* in the Influenza group) and Gram-negative rods (*Escherichia coli* in the COVID-19 group and *Haemophilus influenza* in the Influenza group). Only two COVID-19 patients were diagnosed with a viral co-infection (14%), whereas 14 (35%) were diagnosed in the Influenza group (*p* = 0.181). No co-infections were observed with *Aspergillus* sp. in the COVID-19 group, whereas 8 (20%) were observed in the Influenza group, (*p* = 0.002). Seven of those patients had chronic pulmonary disease, five were receiving chronic steroid treatment, and one was a solid organ transplant recipient. The median time to ICU admission for these patients with invasive aspergillosis infections was 15 days (12–20), which is significantly longer than the median time to ICU admission for the entire cohort: 1.5 days (0–6).

Table 3 provides the results of the univariate logistic regression model exploring the risk factors for co-infections. Appendix A provides the frequencies of each variable among the co-infected and non-co-infected patients in both cohorts. Influenza infection and immunosuppression (IS) therapy were identified as risk factors for co-infections upon ICU admission. Each variable for which the univariate odds ratio (OR) yielded a *p*-value ≤ 0.1 was then included in a multivariable logistic regression model. Only IS therapy was identified as an independent risk factor for co-infections among both the COVID-19 and Influenza patients (OR 6.07 (1.15–35.73), *p* = 0.033, and 9.87 (1.54–197.90), *p* = 0.047, respectively).

### 2.4. Secondary Infections

The secondary infections are presented in Table 4. The COVID-19 patients had significantly more secondary infections than the Influenza patients. The secondary infections were mostly caused by Gram-negative bacilli in both groups. Significantly more secondary infections were caused by multi-resistant bacteria in the COVID-19 group than in the Influenza group. On the other hand, the Influenza patients presented more secondary infections due to *Aspergillus* sp. than the COVID-19 patients (9/29, 31% vs. 1/60, 2%, *p* < 0.001). No viral secondary infections were reported for the first secondary infection in either cohort.

The univariate logistic regression model of the risk factors for secondary infections are reported in Table 5. Appendix A provides the frequencies of each variable in the patients with secondary infections, compared to those without, in both cohorts of patients. Obesity, baseline SOFA scores, and treatment with vasopressors were identified as risk factors for secondary infections in the COVID-19 group, and support with Extra Corporeal Membrane Oxygenation (ECMO) was identified as such in both the COVID-19 and Influenza patients. Each variable for which the univariate OR yielded a *p*-value ≤ 0.1 was included in a multivariable logistic regression model. Treatment with vasopressors was the only independent risk factor identified for secondary infections in the COVID-19 group, with an OR of 16.23 (3.36–100.42; *p* < 0.001) and ECMO was the only independent risk factor identified for secondary infections in the Influenza group, with an OR of 22.8 (3.38–457.75; *p* = 0.006).

### 2.5. Outcome and Risk Factors for ICU Death

The data on the outcomes are reported in Table 6. The COVID-19 patients stayed significantly longer in the ICU and needed non-significantly longer mechanical ventilation than the Influenza patients. ICU mortality was comparable between both groups, with an overall mortality of 43/112 (38%). The risk factors for ICU death were looked for in both cohorts. The factors explored were: sex, age, comorbidities, the Charlson Comorbidity Index, baseline SOFA, baseline SAPS 3, baseline PaO_2_/FiO_2_ ratio upon admission, the use of vasopressors, ECMO, co-infections, secondary infections, the number of secondary infectious events, and infections due to multi-drug resistant bacteria. In the COVID-19 group, obesity, hypertension, diabetes, the Charlson Comorbidity Index, and having a solid organ transplant were identified as risk factors for ICU death in the univariate analysis (Table 7). In both groups, baseline SOFA, SAPS3, and treatment with vasopressors were also identified as risk factors for ICU death. In the multivariable analysis, obesity, arterial hypertension, a high Charlson Comorbidity Index, and treatment with vasopressors were identified as independent risk factors for ICU death in the COVID-19 group, with ORs of 4.71 (1.07–23.54; *p*= 0.44), 4.97 (1.06–30.50; *p* = 0.05), 1.54 (1.13–2.29; *p* = 0.014), and 16.13 (2.02–377.47; *p* = 0.25), respectively. However, no risk factors for death were identified for the Influenza group. Furthermore, neither co-infections, secondary infections, nor the number of secondary infections were identified as risk factors for death in either the COVID-19 or Influenza groups.

## 3. Discussion

In this study, we compared the co and secondary infections in critically ill COVID-19 patients admitted during the first wave of the pandemic in a single Belgian University hospital to those in critically ill Influenza patients. We also looked for risk factors for infectious events and death. To our knowledge, this is the first comparative study conducted specifically on ICU patients. We report fewer co-infections yet more secondary infections in the COVID-19 patients compared to the Influenza ICU patients. However, the time between the hospital arrival and the ICU admission was shorter—and the ICU length of stay was significantly longer—in the COVID-19 patients compared to the Influenza patients. Because our data come from the first pandemic wave and patient management has evolved since this moment, we will compare our results in this discussion to other first wave cohorts.

Although both COVID-19 and Influenza are viral diseases, the clinical presentations upon ICU admission differed between our two cohorts. At presentation, the COVID-19 patients in our cohort presented more ground glass features on a CT-scan and a lower leucocyte count, but they presented a higher lymphocyte count and higher c-reactive protein and lactate dehydrogenase values than the Influenza patients. These elements may help to differentiate between these diseases at the time of admission. Indeed, D’Onofrio et al. tried to identify factors that could help distinguish COVID-19 infection from Influenza infections in patients suspected of sepsis in the emergency department. They also reported a lower leucocyte count and higher lactate dehydrogenase values upon admission in the COVID-19 patients compared to the Influenza patients [17].

Despite similar Charlson scores upon ICU admission in our study, the Influenza patients had more comorbidities, and more of them were immunosuppressed compared to the COVID-19 patients. They were hospitalized for longer in the general ward before being admitted to the ICU, and they received more antibiotics before and during the first 48 h of ICU admission. This may partially explain why the number of co-infections observed among the Influenza group upon ICU admission was almost threefold higher than that observed among the COVID-19 cohort.

One out of five of our COVID-19 patients presented a co-infection, concordant with the prevalence rates of 4.3% to 28% previously reported in the literature [3,4,5,7,9]. However, if we consider only respiratory co-infections, two French studies reported higher rates than what we observed: 19.8% and 27.7% compared to 16%, respectively [6,8]. Half of the Influenza patients presented a bacterial co-infection, which was consistent with the values reported in the literature, varying from 5.9 to 51.1% [12].

The pathogens responsible for the co-infections were mostly bacterial, followed by viruses and fungi in both the Influenza and COVID-19 patients. We found 16% of co-infections with *Aspergillus* spp. in the Influenza group but none in the COVID-19 group. This observation could be due to the more frequent chronic consumption of corticosteroids in the Influenza group compared to the COVID-19 patients. Only IS therapy was identified as an independent risk factor by the multivariate analysis for co-infections in both groups. This risk factor for co-infections is consistent with the Influenza literature [13].

Secondary infections were observed more than two times as frequently in the COVID-19 patients than in the Influenza patients. Indeed, two out of three COVID-19 patients in our cohort presented at least one secondary infection, consistent with other studies on secondary infections in COVID-19 patients. Soriano et al., in a retrospective study, reported a secondary infections rate of 51% in 83 COVID-19 ICU patients [4]. Considering only ventilator-associated pneumonia (VAP), Razazi et al. reported that 64% of patients experienced at least one incidence of VAP within 8 (5–12) days of mechanical ventilation in a retrospective study on 90 patients [7]. This higher number of secondary infections may be partially explained by the longer ICU stay of the COVID-19 patients. The time to the onset of secondary infections was similar in the COVID-19 and Influenza cohorts, with a median time of 7 days. Bardi et al., who described nosocomial infections in COVID-19 ICU patients, found a time to onset of 9 days (IQR 5-11) [9]. Elabbadi et al., who described respiratory co and secondary bacterial infections in ICU patients, found a similar delay of 7.5 days [8]. Kokkoris et al., who described secondary blood stream infections in COVID-19 ICU patients, found a slightly longer time to onset of 11 days [10]. The rate of secondary infections may be more significant in the following waves of COVID-19, as glucocorticoids have become the standard of care [18]. Rothe and al. compared ICU COVID-19 patients treated or untreated with glucocorticoids (second versus first wave of COVID-19 in Germany). In their retrospective study, the use of dexamethasone was associated with more pulmonary infectious complications [19]. However, IS therapy was not identified as a risk factor for secondary infections in our study, possibly due to the small cohort size. Tocilizumab has also become the standard of care for COVID-19 patients, but it was not associated with more secondary infections in a recent meta-analysis [20].

The site of the secondary infections in the Influenza and COVID-19 patients was mostly respiratory, concordant with previous reports [4]. Thirty-five percent of these infections were bacteremia in the COVID-19 group, and the pathogens were mostly gram-negative bacilli, consistent with the literature (30.7–40%) [4,7,8,10]. As for co-infections, we found more secondary infections due to *Aspergillus *spp. in the Influenza group compared to the COVID-19 group. In the literature, Bardi et al. described two ventilator-associated infections due to *Aspergillus fumigatus* and one incidence of hospital-acquired pneumonia in COVID-19 ICU patients [9]. The only prospective multicentric study that evaluated coronavirus-associated pulmonary aspergillosis reported 27.7% of Aspergillosis-related secondary infections, with a median of 4 (2–8) days after ICU admission [21]. Various elements could explain this higher rate of infection compared to our observations: the study included only ARDS patients on mechanical ventilation (for more than 48 h), and most patients had received corticosteroids (60% in the aspergillus group and 46.6% in the non-aspergillus group).

The proportion of infections due to multi-drug resistant bacteria was 32% in the COVID-19 group, similar to the findings published by Bardi et al., who reported that 31% of the observed nosocomial infections in their cohort were due to multidrug resistant microorganisms [9]. The number of infections due to multidrug resistant bacteria was significantly greater in the COVID-19 patients than in the Influenza patients during the first secondary event, but it was similar for the second and third events. These results were not expected given the fact that the Influenza group consumed more antibiotics than the COVID-19 cohort at the time of ICU admission, and previous antibiotic therapy is a recognized risk factor for infections due to the multidrug resistant pathogens in the ICU setting [22].

The COVID-19 patients had a longer ICU stay and spent more time on mechanical ventilation than the Influenza patients, as previously described [15]. Despite these elements, the ICU mortality between the two groups in our cohort was comparable. On the other hand, Piroth et al., who compared COVID-19 patients with 2018–2019 seasonal Influenza patients, found a higher ICU mortality in the COVID-19 group (27.1%) than in the Influenza group (18%) [15]. In the literature, the ICU mortality of first wave COVID-19 patients varied from 24.1% to 36% [4,9]. In our study, neither co nor secondary infections were identified as risk factors for ICU death. Kreitman et al., who prospectively described bacterial respiratory co-infections in COVID-19 patients, also found no differences in terms of mortality between the co-infected and non-co-infected patients [6]. Our data are concordant with these results.

The major study limitations are the small sample size and the retrospective design. The small sample size may partially explain why we did not identify more risk factors for co and secondary infections. The rates of co and secondary infections in our study are concordant with the current literature. However, these rates may be overestimated, as the clinical and biological signs of co and secondary infection, such as fever and CRP, are also elevated in COVID-19 without infectious complications, making the differential diagnosis between simple colonization and infection difficult in this context. This study is nevertheless of interest because it analyzes microbiological events in COVID-19 patients before the introduction of other therapies such as glucocorticoids, convalescent plasma, tociluzimab, jak-inhibitors, etc., It also shows that there are differences between Influenza and COVID-19 in terms of disease presentation.

## 4. Materials and Methods

We performed a retrospective, monocentric study at Erasme Hospital, a 1048-bed teaching hospital (including 32 ICU beds) in Brussels, Belgium. We compared the infectious complications observed in the ICU between confirmed COVID-19 and confirmed flu patients from the 13th of March until the 20th of April 2020 and from the 1st of January 2015 until the 20th of April 2020, respectively.

We included all ICU patients with a positive Reverse Transcriptase Polymerase Chain Reaction (RT-PCR), a rapid antigen detection test (RDT), or a viral culture for Influenza A or B or SARS-CoV-2 on nasopharyngeal (NP) swabs or bronchoalveolar lavage (BAL) prior to or within 48 h of ICU admission. The exclusion criteria were patients younger than eighteen years old and pregnant women.

At our institution, Influenza is actively searched for during the yearly flu epidemic (based on Sciensano, the public health institute’s epidemic curves) in all patients admitted for respiratory symptoms or fever using RDT for Influenza A and B (Influ A + B K-SeT, Coris bioConcept, Gembloux, Belgium), viral PCR (cobas^®^ Influenza A/B and RSV Assay (Roche Molecular Systems, Pleasanton, CA, USA)), and cultures. Viral cultures are performed on confluent Vero (African green monkey kidney), MRC5 (human lung), and LLC-MK2 (rhesus monkey kidney) cells (Vircell, Santa-Fé, Spain) in 24-well or 6-well tissue culture plates (Greiner-Bio One, Frickenhausen, Germany) on all NP swabs performed for Influenza A and B detection and on all BAL samples. A multiplex PCR panel with Influenza A and B is also performed on all BAL specimens and NP swabs [23]. As there were no Influenza patients requiring intensive care at our institution during the first wave of the COVID-19 pandemic, we included ICU Influenza patients from the five previous Influenza seasons to obtain a cohort size similar to that of the cohort of ICU COVID-19 patients.

Furthermore, as of the 4th of March 2020, the beginning of the epidemic in Belgium, NP swabs for SARS-CoV-2 PCR were performed on all patients admitted to our hospital. Repeat SARS-CoV-2 PCRs were performed on NP swabs or BALs in patients with initial negative results for SARS-CoV-2 but with respiratory symptoms or an unexplained fever. Influenza was excluded by either an RDT test and culture on an NP swab or a PCR and culture on BAL in every SARS-CoV-2 suspected patient. The RT-PCR tests for SARS-CoV2 were performed using various commercialized automated PCR systems.

In the ICU, multiple biological samples for the microbiology laboratory are collected systematically from all patients admitted with, or who develop during their stay, clinical (i.e., hypotension, fever, cough, change in consciousness, etc.), biological (i.e., elevated C-reactive protein, elevated leucocyte count, etc.), or radiological signs of infection. The sampling, oriented by the clinical exam, includes NP swabs, sputum, BALs, blood, urine, liquid from surgical drains, swabs, or punctures of purulent lesions to perform direct microscopy and culture. We use a customized TaqMan^®^ array card real-time PCR method, targeting 24 viruses, 8 bacteria, and 2 fungi simultaneously [23] on BAL, along with nasal swabs. The results are considered positive if the threshold cycle is below 35. At the start of this pandemic, because of the limited amount of reagent, we had to limit the multiplex panel to ICU patients who underwent a BAL for respiratory deterioration. On BAL, virus/bacteria, mycobacteria, and fungal cultures are systematically performed, as well as galactomannan detection. Galactomannan is also measured on the blood samples if there is clinical suspicion of invasive aspergillosis infection and systematically (twice a week) in immunocompromised patients at a high risk of developing invasive aspergillosis [24]. Furthermore, ICU patients undergo systematic (twice a week) microbiological sampling, including using nasal swabs to detect methicillin resistant *Staphylococcus aureus*, rectal swabs for the screening of multi-resistant Gram-negative bacteria carriage, and respiratory and drain or catheter samples. A PCR on blood for cytomegalovirus detection is also performed twice a week in immunocompromised patients. Every patient is discussed together with the ICU team, infectious diseases (ID) team, and microbiologists on a bi-weekly basis, completed with further daily consults by the ID specialist, if needed, to decide on the best anti-infectious management.

When patients are hospitalized in the ICU with ARDS, respiratory deterioration is systematically documented with a thoracic CT-scan to distinguish secondary infections from other respiratory complications (i.e., pulmonary embolism, pneumothorax, etc.), along with a BAL (if the patient is intubated), tracheal aspiration, or sputum sample.

### 4.1. Data Collection

The medical records were reviewed by two medical doctors. The data collection included each patient’s demographics, comorbidities, reason for ICU admission, delay between hospital and ICU admission, diagnosis upon ICU admission, clinical, microbiological, and laboratory data, radiological results, and outcome. For demographics, age and sex were recorded. For the comorbidities, we calculated the Charlson Comorbidity Index [25] and recorded the presence of chronic pulmonary disease, IS therapy, solid organ transplantation, cardiovascular disease, hypertension, obesity (defined as a body mass index (BMI) greater than or equal to 30 kg/m²), neoplasia, active smoking, diabetes, and renal insufficiency (defined as a glomerular filtration rate below 60 mL/min using the CKD–EPI equation [26]). For clinical data, the Simplified Acute Physiology score 3 (SAPS 3) [27] and the PaO_2_/FiO_2_ ratio upon admission, along with the Sepsis-related Organ Failure Assessment (SOFA) score [28] during the first 24 h in the ICU, were recorded, as well as the need for supplementary oxygen, mechanical ventilation, ventral decubitus, catecholamines, and ECMO during the ICU stay. For the laboratory analyses (d-dimer, total leucocyte, lymphocyte and platelet counts, creatinine, bilirubin, C-reactive protein, lactate dehydrogenase, and creatinine kinase values), the radiological findings (thoracic CT-scan findings: pulmonary embolism, condensations, micronodules, or round glass) were recorded upon admission and at every clinical deterioration.

We considered a documented infection, the association of clinical symptoms (depending on the site of infection), biological signs of infection (see above), and the presence of a relevant pathogen on the microbiological samples. The presence of pathogens had to be correlated with the decision to treat the event—unless the pathogen was a virus (as the infection was considered to be the cause of the clinical deterioration). The relevant pathogens were defined as described below.

For blood cultures (BC), coagulase-negative *Staphylococci* and *Corynebacterium *spp. were considered relevant only if they were isolated in more than one bottle with similar antibiotic susceptibility profiles and persistent after catheter removal/change. All other pathogens identified in the blood cultures were considered relevant. For urinary tract infections, the patients had to have symptoms or fever, an elevated urine white blood cells count, and a positive urine culture (>10^5^ colony forming units/mL) of no more than two isolated micro-organisms. For the purulent lesions obtained by puncture, all of the pathogens identified by the culture were considered relevant. For surgical drain cultures, only the pathogens identified in the samples taken within the first 24 h of placing the drain were considered relevant. For respiratory samples, we excluded the pathogens belonging to the mouth microbiota. As proposed by Verweij et al., an invasive pulmonary Aspergillosis infection was defined as the presence of pulmonary infiltrates with at least one of the following: a galactomannan index superior to one (on blood or BAL) or a positive BAL culture for *Aspergillus* sp. [29].

An antibiogram is systematically performed on relevant bacterial pathogens. Multi-resistant bacteria were defined as having acquired non-susceptibility to at least one agent in three or more antibiotic classes [4]. If an infection was polymicrobial, even if only one of the bacteria was multi-resistant (as defined above), the patient was considered infected by a multi-resistant pathogen.

The infections were defined as co or secondary infections according to the timing of their diagnosis. Co-infections were those diagnosed from 48 h prior to admission until 48 h after admission. All episodes occurring after the first 48 h of admission were recorded as secondary infections. For each infectious episode, the site, the pathogen(s) responsible for the infection, their resistance profile, and the delay between the admission and the event were recorded. The anti-infectious treatments administered were recorded with a distinction between anti-infectious therapies given before ICU admission, those administered empirically during the first 48 h of ICU admission, and those administered after the first microbiological results. Subsequent anti-infectious therapy given for subsequent infectious episodes was also recorded. For the outcome, the length of the ICU-stay, the 28-day mortality, the ICU, and the hospital mortality were recorded.

### 4.2. Statistics

The discrete variables were presented as numbers (%) and compared using the Chi-square-test or Fisher’s exact test, and the continuous variables were expressed as the median and range if not normally distributed and compared with the Mann–Whitney U-test. A univariate, followed by a multivariate logistic regression model, were built to determine the independent risks factors for co-infections, secondary infections, and ICU deaths for the COVID-19 and the Influenza cohorts. The independent risk factors were identified after multivariate analyses models were derived from a backward stepwise analysis from a “full logistic regression model” including all variables for which the univariate odds ratio (OR) yielded a *p*-value ≤ 0.1. A *p*-value < 0.05 was considered statistically significant. The statistical software R (version 4.0.3) [30] was used.

## 5. Conclusions

In conclusion, we report fewer co-infections yet more secondary infections among critically ill COVID-19 patients from the first wave of the pandemic compared to Influenza ICU patients. Most of the infectious complications were respiratory and of bacterial origin. Invasive aspergillosis infections were only observed in the Influenza patients. Immuno-suppressive therapy was identified as an independent risk factor for co-infections among both COVID-19 and Influenza patients. Only ECMO was identified as an independent risk factor for secondary infections in the Influenza patients, and treatment with vasopressors was identified as an independent risk factor for secondary infections in the COVID-19 cohort. Finally, the mortality rate was not greater for the patients with co or secondary infections among both the Influenza and COVID-19 patients.

Our study provides a good picture of the natural course of the COVID-19 disease in critically ill patients. Today, glucocorticoids are the standard of care, and many patients receive anti-Interleukine-6 therapy [31]. Co-infections, secondary infections, and outcomes may differ significantly among the COVID-19 cohort described in this study. Further studies comparing today’s critically ill COVID-19 patients to Influenza patients in terms of co-infections, secondary infections, and outcomes should be pursued. The risk factors for co and secondary infections in critically ill COVID-19 patients need to be better defined to guide antibiotherapy prescription for these patients. On the other hand, antibiotics should be given quickly to immunosuppressed Influenza patients admitted to the ICU, as bacterial co-infections are frequent in these patients.

## Figures and Tables

**Figure 1 antibiotics-11-00704-f001:**
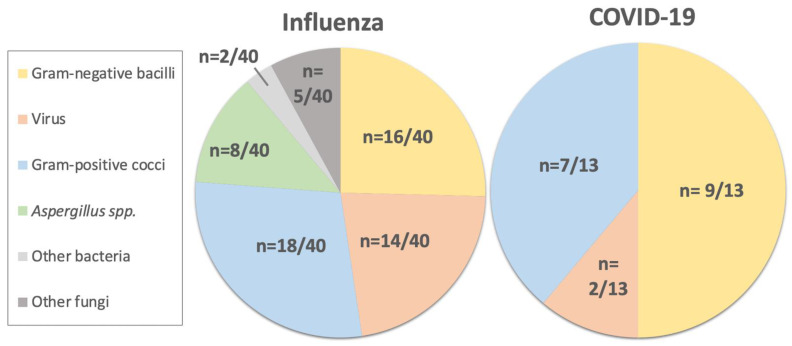
Co-infections in COVID-19 patients versus Influenza ICU patients.

**Table 1 antibiotics-11-00704-t001:** Baseline patient characteristics.

	COVID-19	Influenza	*p*-Value
	N = 57	N = 55	
Sex—male	41 (72)	30 (54)	0.056
Age (years)	61 (53–70)	65 (54–77)	0.108
Comorbidities			
Current smokers, n (%)	6 (10)	23 (42)	<0.001
Obesity	n = 54; 19 (35)	n = 46; 14 (29)	0.534
BMI (kg/m²)	n = 54; 28 (24–31)	n = 46; 28 (24–33)	0.841
Arterial hypertension, n (%)	37 (65)	34 (62)	0.734
Chronic cardiomyopathy, n (%)	9 (16)	23 (42)	0.002
Chronic pulmonary disease, n (%)	17 (30)	27 (49)	0.037
Chronic kidney failure, n (%)	4 (7)	13 (24)	0.014
Diabetes, n (%)	11 (19)	16 (29)	0.226
Active neoplasia, n (%)	3 (5)	3 (5)	0.999
Charlson Comorbidity Index, n (%)	3 (1–5)	4 (3–6)	0.077
Immunosuppressive therapy, n (%)	6 (10)	24 (44)	<0.001
Chronic steroids, n (%)	4 (7)	16 (29)	0.002
Solid organ transplant, n (%)	3 (5)	6 (11)	0.317

Continuous variables are reported as the median (interquartile ranges), and categorical variables are reported as counts (percentages). A *p* < 0.05 was considered statistically significant. Abbreviation—BMI: body mass index.

**Table 2 antibiotics-11-00704-t002:** (a). Clinical data upon ICU admission. (b). Microbiological data upon ICU admission.

**(a)**			
	**COVID-19**	**Influenza**	** *p* ** **-Value**
	**N = 57**	**N = 55**	
Reason for ICU admission			
Respiratory failure, n (%)	51 (89)	47 (86)	0.52
Medical reason, n (%)	6 (11)	7 (13)	0.716
Surgical reason, n (%)	0 (0)	1 (2)	0.477
Hospital to ICU admission, days	1 (0–4)	2 (0–11)	0.012
Baseline SOFA	7 (3–9)	6 (3–10)	0.943
Baseline SAPS 3	55 (46–68)	62 (52–71)	0.07
Baseline PaO₂/FiO₂ ratio	140 (100–194)	152 (112–213)	0.432
CT features	n = 48	n = 27	
Ground glass, n (%)	46 (96)	15 (56)	<0.001
Condensations, n (%)	28 (58)	19 (73)	0.209
Biological data			
Leukocytes, 10³/mm^3^	8.69 (6.52–11.09)	11.03 (8.42–15.00)	0.035
Lymphocytes count, 10³/mm^3^	0.89 (0.57–1.32)	0.54 (0.34–1.04)	0.028
CRP, mg/L	144 (94–230)	63 (41–125)	<0.001
Creatinine, mg/dL	1.1 (0.8–1.6)	1.2 (0.9–1.8)	0.522
LDH, IU/L	506 (360–620)	303 (239–431)	<0.001
Antibiotherapy before ICU admission, n (%)	18 (32)	30 (54)	0.04
Antibiotherapy during the first 48 h, n (%)	22 (39)	44 (80)	<0.001
**(b)**			
	**COVID-19**	**Influenza**	** *p* ** **-Value**
	**N = 57**	**N = 55**	
Bacteriological samples within 48 h of admission			
Total respiratory samples, n (%)	34 (60)	46 (84)	0.005
Sputum or ETA, n (%)	29 (51)	36 (67)	0.091
BAL, n (%)	25 (44)	12 (22)	0.016
Multiplex respiratory PCR panel, n (%)	23 (40)	21 (39)	0.875
Influenza test (PCR or Ag), n (%)	42 (74)	55 (100)	<0.001
Blood cultures, n (%)	54 (95)	36 (66)	<0.001
Urine cultures, n (%)	54 (95)	20 (38)	<0.001
Bacteriological data			
Total co-infections, n (%)	13 (23)	40 (73)	<0.001
Respiratory co-infections, n (%) *	9 (16)	35 (64)	<0.001
Bacteremia, n (%) *	2 (3)	7 (13)	0.091
Urinary tract infection, n (%) *	2 (3)	1 (2)	0.317
Pathogens of documented co-infections **			
Gram-positive coccus, n (%)	7 (54)	18 (45)	0.579
*Staphylococcus aureus,* n (%)	1 (14)	9 (50)	
*Streptococcus pneumoniae,* n (%)	3 (43)	5 (28)	
Other *Streptococcus* spp.*,* n (%)	3 (43)	3 (17)	
Other*,* n (%)	0 (0)	1 (5)	
Gram-negative bacillus, n (%)	9 (69)	16 (40)	0.607
*Escherichia coli*, n (%)	3 (33)	1 (6)	
*Klebsiella* spp., n (%)	1 (11)	4 (25)	
*Pseudomonas aeruginosa,* n (%)	0 (0)	1 (6)	
*Haemophilus influenzae,* n (%)	3 (33)	5 (31)	
Other, n (%)	2 (22)	5 (31)	
Virus, n (%)	2 (14)	14 (35)	0.181
*Adenovirus,* n (%)	2 (100)	0 (0)	
*Cytomegalovirus,* n (%)	0 (0)	4 (29)	
Coronavirus (other than COVID-19), n (%)	0 (0)	4 (29)	
Other, n (%)	0 (0)	8 (57)	
*Aspergillus* sp., n (%)	0 (0)	8 (20)	0.002

Continuous variables are reported as the median (interquartile range), and categorical variables are reported as numbers (percentages). A *p* < 0.05 was considered statistically significant. Abbreviations—ICU: Intensive Care Unit, SOFA: Sequential Organ Failure Assessment, SAPS 3: Simplified Acute Physiology Score 3, PaO₂/FiO₂ ratio: ratio of partial oxygen pressure to the fraction inspired air, CRP: C-reactive protein, LDH: lactate dehydrogenase, ETA: endotracheal aspirations, BAL: bronchoalveolar lavage. Categorical variables are reported as numbers (percentages). A *p* < 0.05 was considered statistically significant. Abbreviations—ETA: endotracheal aspiration, PCR: polymerase chain reaction. * Co-infections could be multisite. ** Co-infections were sometimes polymicrobial.

**Table 3 antibiotics-11-00704-t003:** Univariate analysis of the risk factors for co-infections.

	COVID-19		Influenza	
	n = 57		n = 55	
	OR (CI95%)	*p*-Value	OR (CI95%)	*p*-Value
Sex	0.53 (0.14–2.07)	0.347	1.07 (0.32–3.54)	0.912
Age (years)	0.99 (0.94–1.04)	0.597	0.94 (0.89–0.99)	0.026
Current smokers	1.82 (0.23–10.68)	0.521	2.49 (0.72–10.2)	0.17
Chronic pulmonary disease	1.06 (0.25–3.92)	0.932	0.79 (0.24–2.61)	0.7
Obesity (BMI ≥ 30 kg/m^2^)	1.21 (0.32–4.29)	0.772	0.45 (0.12–1.65)	0.219
Arterial hypertension	0.54 (0.15–1.97)	0.345	0.31 (0.06–1.14)	0.099
Diabetes	*	0.053	0.33 (0.09–1.17)	0.085
Charlson Comorbidity index	1.01 (0.79–1.26)	0.964	0.66 (0.44–0.92)	0.022
Immunosuppressive therapy	6.07 (1.15–35.73)	0.033	6.74 (1.15–128.77)	0.08
Solid organ transplant	1.75 (0.08–19.85)	0.659	2.00 (0.29–40.14)	0.543
Baseline SOFA	1.16 (1–1.38)	0.066	1.01 (0.88–1.18)	0.866
Baseline SAPS 3	1 (0.97–1.04)	0.878	1.00 (0.96–1.04)	0.856
Baseline PaO₂/FiO₂ ratio	1 (0.99–1.01)	0.749	1.00 (0.99–1)	0.153
Influenza			9.03 (3.94–22.02)	<0.001

Univariate r UnU. Univariate regression analysis, except *: Fisher’s exact test. A *p* < 0.05 was considered statistically significant. The data are presented as the odds ratio (OR) with its 95% confidence interval (CI 95%). Abbreviations—BMI: body mass index, *SOFA*: Sequential Organ Failure Assessment, SAPS 3: Simplified Acute Physiology Score 3, PaO₂/FiO₂ ratio*:* ratio of partial oxygen pressure to the fraction inspired air.

**Table 4 antibiotics-11-00704-t004:** Secondary infections.

	COVID-19	Influenza	*p*-Value
	N = 57	N = 55	
Total number of infectious events	60	29	-
Secondary infections			
Event 1, n (%)	37 (65)	16 (29)	<0.001
Time to onset (day)	8 (2–23)	6.5 (0–17)	0.484
Bacteremia, n (%) *	13 (35)	4 (25)	0.538
Respiratory infections, n (%) *	35 (95)	13 (81)	0.155
Others, n (%) *	3 (8)	3 (19)	0.351
Types of pathogens **			
Gram-positive cocci, n (%)	6 (16)	2 (20)	>0.999
Gram-negative bacilli, n (%)	34 (92)	9 (90)	>0.999
Virus, n (%)	0 (0)	0 (0)	>0.999
*Aspergillus* sp., n (%)	1 (3)	7 (44)	<0.001
Event 2, n (%)	18 (32)	7(13)	0.017
Time to onset (day)	18 (11–29)	16 (8–21)	0.048
Bacteremia, n (%) *	4 (22)	2 (29)	>0.999
Respiratory infections, n (%) *	13 (72)	5 (71)	>0.999
Others, n (%) *	2 (11)	0 (0)	>0.999
Types of pathogens **			
Gram-positive cocci, n (%)	3 (18)	0 (0)	>0.999
Gram-negative bacilli, n (%)	16 (94)	4 (100)	>0.999
Virus, n (%)	1 (6)	1 (14)	0.49
*Aspergillus* sp., n (%)	0 (0)	0 (0)	>0.999
Other fungi, n (%)	0 (0)	2 (40)	0.039
Event 3, n (%)	5 (9)	6 (11)	0.704
Time to onset (day)	23 (18–31)	22 (15–69)	0.583
Bacteremia, n (%) *	5 (100)	1 (33)	0.061
Respiratory infections, n (%) *	2 (40)	5 (83)	0.242
Others, n (%) *	0 (0)	0 (0)	>0.999
Types of pathogens **			
Gram-positive cocci, n (%)	2 (50)	1 (20)	0.524
Gram-negative bacilli, n (%)	4 (100)	5 (100)	>0.999
Virus, n (%)	1 (20)	3 (50)	0.546
*Aspergillus* sp., n (%)	0 (0)	2 (33)	0.456
Other fungi, n (%)	0 (0)	2 (33)	0.456

Continuous variables are reported as the median (interquartile range) and categorical variables are reported as numbers (percentages). Time is reported as the median (minimum–maximum). A *p* < 0.05 was considered statistically significant. * Secondary infections could be multisite. ** Secondary infections were sometimes polymicrobial.

**Table 5 antibiotics-11-00704-t005:** Univariate analysis of the risk factors for secondary infections.

	COVID-19 n = 57		Influenza n = 55	
	OR (CI95%)	*p*-Value	OR (CI95%)	*p*-Value
Sex	2.42 (0.73–8.11)	0.146	1.1 (0.34–3.65)	0.871
Age (years)	0.98 (0.94–1.03)	0.472	0.97 (0.93–1.01)	0.139
Current smokers	1.09 (0.19–8.42)	0.924	1.12 (0.34–3.63)	0.852
Chronic pulmonary disease	0.99 (0.31–3.39)	0.983	0.51 (0.15–1.66)	0.274
Obesity	4.82 (1.33–23.19)	0.026	0.82 (0.2–2.96)	0.768
Arterial hypertension	1.39 (0.44–4.31)	0.568	0.72 (0.22–2.41)	0.587
Diabetes	*	0.548	0.46 (0.09–1.75)	0.286
Charlson Comorbidity Index	0.87 (0.7–1.07)	0.196	0.85 (0.61–1.17)	0.32
Immunosuppressive therapy	1.41 (0.27–10.52)	0.701	1.52 (0.39–5.46)	0.529
Solid organ transplant	*	0.545	0.45 (0.02–3.14)	0.487
Baseline SOFA	1.21 (1.04–1.45)	0.018	1.01 (0.86–1.17)	0.922
Baseline SAPS 3	0.99 (0.96–1.02)	0.482	1.03 (0.99–1.07)	0.192
Baseline PaO₂/FiO₂ ratio	0.99 (0.98–0.99)	0.004	1 (0.99–1)	0.425
Vasopressors	19.43 (4.8–104.01)	<0.001	3.92 (0.91–27.31)	0.098
ECMO	*	0.005	22.8 (3.38–457.75)	0.006
Co-infections	0.54 (0.15–1.97)	0.345	3.5 (0.81–24.45)	0.131
Influenza			0.22 (0.1–0.48)	<0.001

Univariate regression analysis, except *: Fisher’s exact test. A *p* < 0.05 was considered statistically significant. The data are presented as the odds ratio (OR) with its 95% confidence interval (CI 95%). Abbreviations—BMI: body mass index, SOFA: Sequential Organ Failure Assessment, SAPS 3: Simplified Acute Physiology Score 3, PaO₂/FiO₂ ratio: ratio of partial oxygen pressure to the fraction inspired air, ECMO: extracorporeal membrane oxygenation.

**Table 6 antibiotics-11-00704-t006:** ICU Patient outcome.

	COVID-19	Influenza	*p*-Value
	N = 57	N = 55	
Supportive measures			
Optiflow, n (%)	8 (14)	15 (27)	0.083
NIV/CPAP, n (%)	32 (56)	31 (56)	0.981
Invasive mechanical ventilation, n (%)	41 (72)	36 (65)	0.591
Prone positioning, n (%)	35 (61)	9 (16)	<0.001
ECMO, n (%)	13 (23)	7 (13)	0.164
Vasopressors, n (%)	41 (72)	39 (71)	0.905
Renal replacement therapy, n (%)	16 (28)	9 (16)	0.137
ICU stay (days)	15 (4–29)	5 (2–10)	0.001
Mechanical ventilation duration (days)	11 (0–22)	4 (1–18)	0.716
ICU mortality (%), n (%)	20 (35)	23 (41)	0.464

Continuous variables are reported as the median (interquartile range), and categorical variables are reported as numbers (percentages). A *p* < 0.05 was considered statistically significant. Abbreviations—ICU: intensive care unit, NIV: non-invasive ventilation, CPAP: continuous positive airway pressure, ECMO: extracorporeal membrane oxygenation.

**Table 7 antibiotics-11-00704-t007:** Univariate analysis of the risk factors for ICU mortality.

	COVID-19 n = 57		Influenza n = 55	
	OR (CI95%)	*p*-Value	OR (CI95%)	*p*-Value
Sex	1.92 (0.556–7.81)	0.323	1.56 (0.529–0.72)	0.426
Age (years)	1.001 (0.764–1.05)	0.745	101 (0.975–1.05)	0.563
Current smokers	0.917 (0.119–5.18)	0.924	0.438 (0.136–1.32)	0.15
Chronic pulmonary disease	0.29 (0.06–1.06)	0.082	0.5 (0.164–1.47)	0.213
Obesity (BMI ≥ 30 kg/m^2^)	5.44 (1.71–18.77)	0.005	0.958 (0.273–32)	0.945
Arterial hypertension	4.82 (1.33–23.19)	0.026	0.496 (0.16–1.49)	0.215
Diabetes	4.44 (1.15–19.48)	0.035	0.051 (0.003–0.293)	0.006
Charlson Comorbidity Index	1.37 (1.1–1.8)	0.012	0.858 (0.648–1.17)	0.264
Immunosuppressive therapy	2 (0.333–11.84)	0.425	0.533 (0.171–1.58)	0.264
Solid organ transplant		0.012 *	0.245 (0.012–1.67)	0.215
Baseline SOFA	1.22 (1.05–1.44)	0.012	1.18 (1.035–1.38)	0.019
Baseline SAPS	1.04 (1.001–1.09)	0.038	1.06 (1.02–1.11)	0.011
Baseline PaO₂/FiO₂ ratio	0.998 (0.991–1.004)	0.463	0.988 (0.977–0.996)	0.008
Vasopressors	12.95 (2.29–245.19)	0.018	4.56 (1.24–22.22)	0.034
ECMO	3.45 (0.936–13.62)	0.065	2.04 (0.406–11.3)	0.386
Co-infections	0.778 (0.187–2.87)	0.711	4 (1.08–19.55)	0.054
Secondary infections	3.05 (0.91–12.24)	0.087	1.60 (0.49–5.26)	0.432
Number of surinfections	1.415 (0.825–2.5)	0.211	1.14 (0.685–1.9)	0.606
Drug multiresistance bacteria	*	>0.999 *	0.463 (0.021–3.95)	0.519
Influenza			1.32 (0.62–2.87)	0.465

Univariate regression analysis, except *: Fisher’s exact test. *p* < 0.05 was considered statistically significant. The data are presented as the odds ratio (OR) with its 95% confidence interval (CI95%). Abbreviations—BMI: body mass index, SOFA: Sequential Organ Failure Assessment, SAPS 3: Simplified Acute physiology Score 3, PaO₂/FiO₂ ratio: ratio of partial oxygen pressure to the fraction inspired air, ECMO: extracorporeal membrane oxygenation.

## Data Availability

The data presented in this study are available on request from the corresponding author. The data are not publicly available due to the medical nature of this data, even if the data is anonymized.

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
