# Peer review of "A Retrospective, Monocentric Study Comparing Co and Secondary Infections in Critically Ill COVID-19 and Influenza Patients"

_antibiotics, 2022, doi:10.3390/antibiotics11060704_

Round 1

Reviewer 1 Report

The comparison of co- and secondary infections in patients with COVID-19 and influenza is described well and informative.

Some comments should be addressed to improve this article. 

  1. 2.3. Co-infections & 2.4. Secondary infections: Please provide the specific species of infected pathogens and the proportions of them in the revised supplementary tables.
  2. The criteria for selected variables for multivariate analysis have to be described. 
  3. Descriptions for risk factors of invasive mechanical ventilation and longer ICU stays as outcomes  are recommended.

Author Response

1/ 2.3. Co-infections & 2.4. Secondary infections: Please provide the specific species of infected pathogens and the proportions of them in the revised supplementary tables.

Thank you for this suggestion. We have divided the table 2 into table 2a and table 2b. We have added more detailed information concerning specific species in Table 2b.

2/ The criteria for selected variables for multivariate analysis have to be described.

We believe that this is already described.  Indeed, in the materials and methods, it is written, “Independent risk factors were identified after multivariable analyses models were derived from a backward stepwise analysis from a “full logistic regression model” including all variables for which the univariate odds ratio (OR) yielded a p-value ≤ 0.1.”

3/ Descriptions for risk factors of invasive mechanical ventilation and longer ICU stays as outcomes  are recommended.

Thank you for this suggestion. However, our study concerns the first wave of the COVID 19 pandemic., During this first wave, the ICUs became rapidly saturated, and almost all COVID-19 patients who remained in the ICU were intubated. As the pandemic progressed, we applied more non-invasive ventilation to patients than at the very beginning of the pandemic. Risk factors for invasive mechanical ventilation and length of ICU stay were partly influenced by the saturation of ICUs. Therefore, we do not believe that there is much interest in identifying risk factors for length of ICU stay and need for mechanical ventilation. Mortality, on the other hand, is a hard end point, which is why we have provided risk factors for death.

Reviewer 2 Report

This manuscript needs some improvement before publication. In abstract section Jan, 2015 -April 20, 2020? COVID was not in 2015, they how they did this study? In table 2  pathogens of documented co-infection section % is not matching with No. so correct it. In line 114-115,  co-infection were predominantly respiratory in both groups but Tabe2 shoeing this predominantly in influenza but not in COVID, correct it. Correct the percentage and No. of table 4.

Author Response

1/ In abstract section Jan, 2015 -April 20, 2020? COVID was not in 2015, they how they did this study?

Thank you for this comment. We believe that there has been some misunderstanding. Of course, we only included COVID-19 patients in our study since the start of the pandemic, not before.

We have modified this sentence to be clearer.

2/ In table 2 pathogens of documented co-infection section % is not matching with No. so correct it.

Thank you for the thorough review of the article. We have added a comment as a table footprint explaining that some infections were polymicrobial.  We also corrected the table.

3/ In line 114-115,  co-infection were predominantly respiratory in both groups but Tabe2 shoeing this predominantly in influenza but not in COVID, correct it.

We think that there is a misunderstanding. We confirm that our data shows in Table 2 that in the COVID-19 cohort, respiratory co-infections were also predominant (16% vs 3% of bacteriemias and 3% of urinary tract infections).

4/ Correct the percentage and No. of table 4.

A comment has been added as a table footprint. Explaining that secondary infections could be polymicrobial and that urinary tract infections or respiratory infections could be associated with a bacteriemia.

Reviewer 3 Report

The manuscript provides an interesting insight into co-infections and secondary infections in critically ill COVID- 19 and influenza patients in Belgium. The paper is clear and detailed. However, some discrepancies emerge between the tables and the description of the tables in the text. A revision of punctuation and definitions would be recommended in some cases.

Abstract

  • A revision of punctuation would be advisable, in particular avoiding excessive use of round brackets (line 21; line 25).

Results

  • In Table 1 it would be better to replace “mean age” variable with “age” (in the legend it is specified that it is a median).
  • Avoid excessive use of round brackets (line 186).
  • With reference to Table 5 it would be advisable to define the analysis as univariate or multivariable logistic regression model and not as “multivariate analysis” (line199; line 203).
  • Line 201 describes an OR of 16.23 [3.36 – 100.42], p<0.001. In which table is this value shown?
  • Section 2.5 refers to a univariate analysis whose data are not shown (line223; line 225). Why is the data not shown?

Discussion

The challenge related to the diagnosis of pneumonia in COVID-19 patients should be added.

Were patients discharged from the ICU still under surveillance? If not, it should be added as a limitation.

Materials and Methods

With reference to patient recruitment, it would be advisable to better explain why patients with Influenza were recruited between 2015 and 2020.

Line 478. Multivariate or multivariable?

Conclusions

  • “Only a high Charlson comorbidity index was identified as an independent risk factor for secondary infections in the Influenza patients” (line 490). Table 5 shows that the OR of the Charlson comorbidity index is not significant, does this sentence refer to ECMO instead of Charlson comorbidity index?
  • “Treatment with vasopressors was identified as an independent risk factor for secondary infections in the COVID-19 cohort” (line 493). Table 5 shows that obesity, baseline SOFA scores, treatment with vasopressors and ECMO were identified as risk factors for secondary infections in the COVID-19 group (not only vasopressors) therefore it would be appropriate to list all the risk factors.

Figure 1

The resolution of this image should be improved.

Author Response

1/ A revision of punctuation would be advisable, in particular avoiding excessive use of round brackets (line 21; line 25).

We have done this.

2/ In Table 1 it would be better to replace “mean age” variable with “age” (in the legend it is specified that it is a median).

Thank you for pointing this mistake out. We have made the correction.

3/ Avoid excessive use of round brackets (line 186).

 We have made the appropriate modifications.

4/ With reference to Table 5 it would be advisable to define the analysis as univariate or multivariable logistic regression model and not as “multivariate analysis” (line199; line 203).

Thank you very much for this comment. We have made the appropriate modifications.

5/ Line 201 describes an OR of 16.23 [3.36 – 100.42], p<0.001. In which table is this value shown?

Because only a few variables were eligible for the multivariable logistic regression, and only one result was statistically significant, we decided to describe the results and not present this result in a  table.

6/ Section 2.5 refers to a univariate analysis whose data are not shown (line223; line 225). Why is the data not shown?

A new table (Table 7) has now been added with the data.

7/ The challenge related to the diagnosis of pneumonia in COVID-19 patients should be added.

Thank you for this important remark. We have added a sentence in the limitations section of the article.

8/ Were patients discharged from the ICU still under surveillance? If not, it should be added as a limitation.

Patients were still under surveillance in dedicated COVID sections of the hospital when they were discharged from the ICU. However, no systematic microbiological samples were taken in these other sectors of the hospital. We have chosen to not add this point to the limitations of our study because we believe that this information is out of scope of our study. Indeed, we describe the co-infections and secondary infections in the cohorts during their hospitalization in the ICU.

9/ With reference to patient recruitment, it would be advisable to better explain why patients with Influenza were recruited between 2015 and 2020.

Because of the small number of Influenza patients needing admission to the intensive care each year, we had to include Influenza patients admitted to our ICU over the last 5 years to obtain an Influenza cohort similar in size to the COVID-19cohort. A sentence has been added to the material and methods section.

10/ Line 478. Multivariate or multivariable?

Thank you very much for this remark. We have corrected the article to “multivariable”.

11/“Only a high Charlson comorbidity index was identified as an independent risk factor for secondary infections in the Influenza patients” (line 490). Table 5 shows that the OR of the Charlson comorbidity index is not significant, does this sentence refer to ECMO instead of Charlson comorbidity index?

Yes, exactly, thank you! We have made the appropriate modifications.

12/ “Treatment with vasopressors was identified as an independent risk factor for secondary infections in the COVID-19 cohort” (line 493). Table 5 shows that obesity, baseline SOFA scores, treatment with vasopressors and ECMO were identified as risk factors for secondary infections in the COVID-19 group (not only vasopressors) therefore it would be appropriate to list all the risk factors.

Thank you for this comment. We agree with you, and have made the appropriate modifications.

13/ Figure 1 - The resolution of this image should be improved

This figure has been updated. We went from a dpi of 144 to 600.

Round 2

Reviewer 2 Report

Authors have responded very well of all my criticism.